# Importance Sampling for Cost-Optimized Estimation of Burn Probability Maps in Wildfire Monte Carlo Simulations

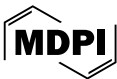

**Valentin Waeselynck** [1,*] and **David Saah** [2]

1   Spatial Informatics Group, Pleasanton, CA 94566, USA
2   Department of Environmental Science, University of San Francisco, San Francisco, CA 94117, USA; dssaah@usfca.edu
*   Correspondence: vwaeselynck@sig-gis.com

**Abstract:** Background: Wildfire modelers rely on Monte Carlo simulations of wildland fire to produce burn probability maps. These simulations are computationally expensive. Methods: We study the application of importance sampling to accelerate the estimation of burn probability maps, using L2 distance as the metric of deviation. Results: Assuming a large area of interest, we prove that the optimal proposal distribution reweights the probability of ignitions by the square root of the expected burned area divided by the expected computational cost and then generalize these results to the assets-weighted L2 distance. We also propose a practical approach to searching for a good proposal distribution. Conclusions: These findings contribute quantitative methods for optimizing the precision/computation ratio of wildfire Monte Carlo simulations without biasing the results, offer a principled conceptual framework for justifying and reasoning about other computational shortcuts, and can be readily generalized to a broader spectrum of simulation-based risk modeling.

**Keywords:** Monte Carlo simulations; wildfire; risk modeling; burn probability; variance reduction; sampling error





## 1. Introduction

Monte Carlo simulations [1] are an approach to studying random processes by drawing samples from them. This approach has often been applied to studying the potential impacts of wildfire [2]: burn probability maps can be produced from Monte Carlo simulations by averaging the burned areas of many randomly ignited fires simulated by a wildfire behavior model [3].

Simulating a large number of fires is required to ensure that the Monte Carlo estimator converges to an asymptotic value; otherwise, running the simulations again may produce a significantly different result. This random variability is known as the *variance* of the Monte Carlo estimator. Reducing the variance is required for making the estimation precise.

Importance sampling is a general strategy for reducing the computational cost of Monte Carlo simulations. Importance sampling redistributes the sampling probability of simulation outcomes while inversely scaling their contribution to simulation results. This preserves the expected value of the Monte Carlo estimator while altering its variance, enabling faster convergence to the asymptotic estimand. In the case of wildfire simulations, importance sampling boosts or depresses the probability that some fires will be simulated and accordingly decreases or increases their contribution to the aggregated simulation results. This process involves choosing a *proposal distribution* from which to draw the samples; choosing the proposal distribution well is the key to successfully implementing importance sampling. In [1], importance sampling is described as "the most fundamental variance reduction technique". Therefore, it appears important for fire modelers to consider using importance sampling in their simulations; this article aims to help them understand its underlying principles, assess its potential benefits, and implement it when they are significant.

This paper is structured as follows. Section 3 develops the conceptual methods introduced by this paper. This mostly consists of framing the problem of computational efficiency in wildfire simulations through concepts like the *cost-to-precision ratio* and *efficiency gain*; introducing a generalized notion of variance (based on the $L_2$ norm) suited to estimating burn probability maps (Section 3.3, Equation (2)); and then recalling the definition of importance sampling (Section 3.4) as a technique for accelerating estimator convergence in Monte Carlo simulations, adapting it to the context of wildfire simulations with $L_2$ variance, calculating its performance characteristics, considering various generalizations, and suggesting approaches to implementing the approach (Section 3.12). Section 4 illustrates these methods with various case studies. Section 5 discusses the limitations of this approach. Table 1 summarizes the symbols most commonly encountered in formulas; their meaning is defined in Section 3.

**Table 1.** Table of symbols.

| Symbol | Type | Meaning |
|---|---|---|
| $\mathbb{E}[\hat{z}]$ | Same as $\hat{x}$ | Expected value of random [1] variable $\hat{z}$ |
| $\text{Var}[\hat{z}]$ | $\mathbb{R}$ | Variance of $\hat{z}$ |
| $\text{Prec}[\hat{z}]$ | $\mathbb{R}$ | Precision (inverse variance) of $\hat{z}$ |
| $\mathcal{A}$ | | Area of interest |
| $s$ | $s \in \mathcal{A}$ | Spatial location |
| $F$ | | (Potentially) simulated fire |
| $\hat{F}$ | | Random variable representing the natural distribution of simulated fires ($\hat{F} \sim p(\cdot)$) |
| $a(F)$ | $\mathbb{R}$ | Area (or value) burned by fire $F$ |
| $b_F$ | $\mathcal{A} \to \{0,1\}$ | Binary burn map [2] (locations burned by $F$) |
| $c(F)$ | $\mathbb{R}$ | Computational cost |
| $\mu$ | $\mathcal{A} \to \mathbb{R}$ | Asymptotic burn probability map (estimation target) |
| $\|z\|$ | $\mathbb{R}$ | $L_2$ norm of map $\|z\|$ |
| $\|z\|_u$ | $\mathbb{R}$ | Impact-weighted $L_2$ norm of map $\|z\|$ |
| $u(\cdot)$ | $\mathcal{A} \to \mathbb{R}$ | Spatial density of (asset) value |
| $\hat{I}$ | | Ignition or initial conditions |
| $p(\cdot)$ | $\mathbb{R}$ | Probability density of the natural distribution |
| $q(\cdot)$ | $\mathbb{R}$ | Probability density of a proposal distribution |
| $w_q(\cdot)$ | $\mathbb{R}$ | Importance weight (ratio $p(\cdot)/q(\cdot)$) under proposal distribution $q$ |
| $\hat{F}^{(q)}$ | | Random variable representing the proposal distribution ($\hat{F}^{(q)} \sim q(\cdot)$) |
| $\hat{m}_q$ | $\mathcal{A} \to \mathbb{R}$ | Importance sampling estimator with proposal distribution $q$ |
| $r_{C/P}^{(q)}$ | $\mathbb{R}$ | Cost-to-precision ratio (inverse efficiency) |
| $g_q$ | $\mathbb{R}$ | Efficiency gain: $g_q := r_{C/P}^{(p)}/r_{C/P}^{(q)}$ |
| $q_c^*(\cdot)$ | $\mathbb{R}$ | Optimal proposal distribution |

[1] Variables with a hat (like $\hat{x}$) are subject to Monte Carlo randomness. [2] $\mathcal{X} \to \mathcal{Y}$ denotes the set of functions from set $\mathcal{X}$ to set $\mathcal{Y}$.

The main contributions of this paper are the following. First, we note that the usual presentations of importance sampling are not directly applicable to wildfire simulations, due to the computational costs of simulated fires being highly imbalanced, and to simulations producing maps rather than scalars. This leads us to introduce a multidimensional notion of variance (Section 3.3) and to motivate concepts like the *cost-to-precision ratio* and *efficiency gain* as relevant metrics of computational efficiency. We then derive optimality bounds that quantify the maximum efficiency gains that can be achieved through importance sampling (Equations (26), (27) and (33)); these equations are remarkably simple thanks to the use of the L2 metric, and in particular, they require knowing only two scalar variables about the target distribution of simulated fires (size and computational cost). More concretely, optimal gains are achieved by simply reweighting the sampling probability of each fire by a factor proportional to its size-to-cost ratio. The upper bound on efficiency gain cannot be computed but is easy enough to estimate based on a small calibration sample of simulated fire; however, it is not trivial to find a proposal distribution that gets close to this

upper bound, as that essentially requires predicting what the size and cost of a fire will be without spending the computational cost of simulating it. This is a very typical problem in statistics—optimal estimation requires perfect knowledge of the sampling distribution, which is what we are trying to estimate—which we propose to address in the usual way: by approximating the sampling distribution using a sample of data along with parametric models. In our context, this means drawing a small calibration sample of simulated fires, from which we derive an objective function that approximates the true efficiency gain, thus enabling the application of optimization algorithms that search for a good proposal distribution; this approach is theorized in Section 3.12 and illustrated in Section 4. Finally, a nontrivial contribution of this paper is to discuss nontrivial pitfalls and limitations of this embodiment of importance sampling, such as creating geographic inequalities in estimation precision (Section 3.13).

## 2. Related Work

This paper is in the context of wildfire simulations applied to risk prediction, as exemplified by [3–5]. Fire behavior simulations are not the only approach to predicting wildfire risk [6]: many authors model fire risk based on statistical models or machine learning methods [7,8]; recent examples are [9,10]. Wildfire simulations are also used for other applications than producing burn probability maps or than risk prediction [2].

Importance sampling has been used for sub-aspects of wildfire simulation. Ref. [11] has studied the use of importance sampling for data assimilation. Ref. [12] uses importance sampling as part of the simulation of spotting behavior. We know of no academic study of importance sampling for accelerating the estimation of wildfire risk maps. We were only half surprised by this absence of prior work: indeed, our anecdotal experience suggests that the fire simulation community has historically had little contact with computational statistics, an issue which this paper aims to help address.

## 3. Materials and Methods

This analysis is conducted in the context of estimating burn probability maps over a large area of interest $\mathcal{A}$.

### 3.1. Background: Mean, Variance, and Precision of Random Maps

For measuring the difference between multidimensional outputs like burn probability maps, we use the $L_2$ norm $\|z\|$, defined by the spatial integral:

$$\|z\|^2 := \int_{s \in \mathcal{A}} |z(s)|^2 ds \tag{1}$$

The distance between two maps $z_1$ and $z_2$ is then $\|z_1 - z_2\|$.

We generalize the notion of variance from random numbers to random maps as follows. If $\hat{z}$ is a random map, then we define the variance of $\hat{z}$ to be the following:

$$\mathrm{Var}[\hat{z}] := \mathbb{E}\left[\|\hat{z} - \mathbb{E}[\hat{z}]\|^2\right] = \mathbb{E}\left[\|\hat{z}\|^2\right] - \|\mathbb{E}[\hat{z}]\|^2 \tag{2}$$

Note that this can also be expressed in terms of pointwise variances—for each spatial location $s$, $\hat{z}(s)$ is a random number of variance $\mathrm{Var}[\hat{z}(s)]$, and we have

$$\mathrm{Var}[\hat{z}] = \int_{s \in \mathcal{A}} \mathrm{Var}[\hat{z}(s)] ds \tag{3}$$

Many properties of scalar variance are retained: variance is additive on independent variables, and a random variable is constant if and only if its variance is zero.

A useful related concept is the *precision*, defined as the inverse variance:

$$\text{Prec}[\hat{z}] := \frac{1}{\text{Var}[\hat{z}]} \tag{4}$$

Aiming for low variance is equivalent to aiming for high precision.

### 3.2. Mean Estimation from Monte Carlo Simulations

Assume that we want to estimate the unknown expectation $\mu := \mathbb{E}[\hat{y}]$ of a distribution of interest, represented by a random map $\hat{y}$. (Note that $\mu$ itself is a map, not a number.) When all we know about $\hat{y}$ is how to draw samples from it, a Monte Carlo simulation lets us estimate $\mu$ by drawing many i.i.d samples $\hat{y}_k \sim \hat{y}$ and computing their empirical mean $\hat{m}_y$:

$$\hat{m}_y := \frac{1}{N} \sum_{k=1}^{N} \hat{y}_k \tag{5}$$

Expectation being linear, it is easily seen that $\mathbb{E}[\hat{m}_y] = \mathbb{E}[\hat{y}] = \mu$. The point of drawing a large number of samples $N$ is to increase the precision of that estimation so that we can rely on $\hat{m}_y \approx \mu$. Indeed, from the properties of variance, the variance and precision of $\hat{m}_y$ are related to the sample size $N$ as follows:

$$\text{Var}[\hat{m}_y] = \frac{1}{N^2} \sum_{k=1}^{N} \text{Var}[\hat{y}_k] = \frac{1}{N} \text{Var}[\hat{y}]$$

$$\text{Prec}[\hat{m}_y] = N\text{Prec}[\hat{y}] \tag{6}$$

The precision formula is insightful: we can think of each sample contributing $\text{Prec}[\hat{y}]$ to estimator precision. Thus, $\text{Prec}[\hat{y}]$ can be viewed as a *precision per sample* performance characteristic of our sampling distribution.

Assume now that with each sample $\hat{y}_k$ is associated a *computational cost* $\hat{c}_k^{(y)}$ of drawing it—typically, $\hat{c}_k^{(y)}$ is expressed in computer time. For example, in wildfire simulation, large fires are often more costly to simulate than small fires. Then, the cost of drawing a large number $N$ of i.i.d samples will be approximately $N\mathbb{E}\left[\hat{c}^{(y)}\right]$. This motivates the following concept of the ***cost-to-precision ratio*, which quantifies the overhead of our estimator:**

$$r_{C/P}[\hat{y}] = \frac{\mathbb{E}\left[\hat{c}^{(y)}\right]}{\text{Prec}[\hat{y}]} = \mathbb{E}\left[\hat{c}^{(y)}\right] \text{Var}[\hat{y}] \tag{7}$$

### 3.3. Burn Probability Maps from Monte Carlo Simulations

We assume the following structure for how simulations run. Each simulation $k$ draws initial conditions $\hat{I}_k$ (typically, $\hat{I}_k$ encompasses the space–time location of ignition and the surrounding weather conditions), from which a fire $\hat{F}_k$ is simulated to yield a binary burn map $b_{\hat{F}_k}$, in which

$$b_{\hat{F}_k}(s) = \begin{cases} 1 & \text{if } \hat{F}_k \text{ burned location } s \\ 0 & \text{otherwise} \end{cases}$$

Thus, each $b_{\hat{F}_k}(s)$ is a Bernoulli random variable, and its expectation $\mu(s) := \mathbb{E}\left[b_{\hat{F}_k}(s)\right]$ is the probability that one simulation burns location $s$.

We assume that the $\hat{F}_k$ are sampled i.i.d. and so will write $\hat{F}$ to denote a random variable representing their common distribution. The same is true for the $\hat{I}_k$.

The goal of the simulations is **to estimate the expected burn probability,** which is defined as the expectation of the burn map, therefore yielding a map of burn probabilities:

$$\mu := \mathbb{E}[b_{\hat{F}}] \tag{8}$$
$$\mu(s) = \mathbb{P}\big[b_{\hat{F}}(s) = 1\big]$$

That estimation is typically performed by using the Crude Monte Carlo (CMC) estimator $\hat{m}_{\mathrm{CMC}}$, which draws a large number $N$ of fires and computes the average of the $b_{\hat{F}_k}$:

$$\hat{m}_{\mathrm{CMC}} := \frac{1}{N} \sum_{k=1}^{N} b_{\hat{F}_k} \tag{9}$$

Because each $b_{\hat{F}}(s)$ is a Bernoulli random variable (taking only 0 and 1 as values), $\mathrm{Var}\big[b_{\hat{F}}\big]$ has an interesting expression in terms of expected fire size:

$$\mathrm{Var}\big[b_{\hat{F}}\big] = \mathbb{E}\Big[\|b_{\hat{F}}\|^2\Big] - \|\mu\|^2 = \mathbb{E}\big[a(\hat{F})\big] - \|\mu\|^2 \tag{10}$$

in which $a(\hat{F}) \geq 0$ is the (random) fire size (say, in m$^2$ or acres, or whatever unit was chosen for $ds$). More precisely, $a(\hat{F})$ is the area burned by the random fire within the AOI $\mathcal{A}$.

The above derivation relies on $\|b_{\hat{F}}\|^2 = a(\hat{F})$, which we now prove. Denoting $\hat{\mathcal{B}} \subseteq \mathcal{A}$ the (random) set of locations burned by a simulated fire, and observing that $|b_{\hat{F}}(s)|^2 = b_{\hat{F}}(s)$, we have the following:

$$\|b_{\hat{F}}\|^2 = \int_{s \in \mathcal{A}} |b_{\hat{F}}(s)|^2 ds = \int_{s \in \mathcal{A}} b_{\hat{F}}(s) ds = \int_{s \in \hat{\mathcal{B}}} 1 ds + \int_{s \notin \hat{\mathcal{B}}} 0 ds = a(\hat{F}) + 0 \tag{11}$$

Another insightful relationship is that the expected fire size is the spatial integral of the expected burn probability:

$$\mathbb{E}\big[a(\hat{F})\big] = \int_{s \in \mathcal{A}} \mu(s) ds \tag{12}$$

Finally, we note a useful approximation. In the common settings where the burn probabilities $\mu(s)$ are low (each location $s$ is burned only by a small minority of simulated fires), then $\mu(s) \ll 1$ implies that $\mathrm{Var}[b_{\hat{F}}(s)] = \mu(s)(1 - \mu(s)) \approx \mu(s) \gg \mu(s)^2$, and since $\mu(s) = \mathbb{E}\big[|b_{\hat{F}}(s)|^2\big]$, it follows that $\|\mu\|^2 \ll \mathbb{E}\big[a(\hat{F})\big]$, and

$$\mathrm{Var}\big[b_{\hat{F}}\big] \approx \mathbb{E}\big[a(\hat{F})\big] \gg \|\mu\|^2 \tag{13}$$

In other words, **the variance of the binary burn map is well approximated by the expected fire size.**

### 3.4. Using Importance Sampling for Reducing the Cost-to-Precision Ratio

We now borrow from the concept of *importance sampling* from the generic Monte Carlo literature [1]. This enables us to design other Monte Carlo estimators for the burn probability $\mu$, with a lower cost-to-precision ratio than the CMC estimator, thus requiring less computation to achieve convergence.

Denote $p(F)$ the "natural" probability distribution from which simulated fires are sampled (i.e., $\hat{F} \sim p$), which is hopefully representative of real-world fire occurrence. The CMC estimator consists of drawing samples $\hat{F}_k$ from $p$ and then averaging the $b_{\hat{F}_k}$ maps. Importance sampling modifies this procedure by drawing samples $\hat{F}_k^{(q)}$ from another

distribution $q(F)$ (called the *proposal distribution*) and reweighting the samples to remain representative of $p$, yielding the *Likelihood Ratio estimator* $\hat{m}_q$:

$$\hat{m}_q := \frac{1}{N} \sum_{k=1}^{N} w_q(\hat{F}_k^{(q)}) b_{\hat{F}_k^{(q)}} \tag{14}$$

in which we defined the *importance weight* as the ratio of probabilities:

$$w_q(F) = \frac{p(F)}{q(F)} \tag{15}$$

It is easily verified that this change of estimator preserves the expected value:

$$\begin{aligned} \mathbb{E}[\hat{m}_q] &= \mathbb{E}\left[ \frac{p(b_{\hat{F}^{(q)}})}{q(b_{\hat{F}^{(q)}})} b_{\hat{F}^{(q)}} \right] \\ &= \int_F dF \, q(F) \frac{p(F)}{q(F)} b_F \\ &= \int_F dF \, p(F) b_F \\ &= \mathbb{E}[b_{\hat{F}}] = \mu = \mathbb{E}[\hat{m}_{\mathrm{CMC}}] \end{aligned} \tag{16}$$

A minor note: we disapprove of the term "Likelihood Ratio estimator", and we find it confusing because it is essentially a ratio of probabilities, not likelihoods; the term "Likelihood Ratio" arises in other parts of statistics (for which it is an apt name, e.g., "Likelihood Ratio test"), but the similarity of the Likelihood Ratio estimator to these is only superficial; thus, we follow the convention of using the term "Likelihood Ratio estimator", but we feel compelled to warn readers of its potential for confusion.

What can make this new estimator valuable is its variance:

$$\begin{aligned} \mathrm{Var}[\hat{m}_q] &= \frac{1}{N} \left( \mathbb{E}\left[ w_q(\hat{F}^{(q)})^2 \|b_{\hat{F}^{(q)}}\|^2 \right] - \|\mu\|^2 \right) \\ &= \frac{1}{N} \left( \mathbb{E}\left[ w_q(\hat{F}) \|b_{\hat{F}}\|^2 \right] - \|\mu\|^2 \right) \end{aligned} \tag{17}$$

The approach of importance sampling consists of seeking a proposal distribution $q$, which minimizes the cost-to-precision ratio $r_{C/P}^{(q)} := r_{C/P}\left[ \frac{p(\hat{F}^{(q)})}{q(\hat{F}^{(q)})} b_{\hat{F}^{(q)}} \right]$. When the cost $\hat{c}$ is constant, this amounts to minimizing variance, in which case a classical result of the Monte Carlo literature says that the optimal proposal distribution $q_{\mathrm{MinVar}}^*$ and the corresponding variance are given by the following:

$$q_{\mathrm{MinVar}}^*(F) \propto \|b_F\| p(F)$$
$$= \mathbb{E}[\|b_{\hat{F}}\|]^{-1} \|b_F\| p(F)$$
$$\mathrm{Var}\left[ \hat{m}_{q_{\mathrm{MinVar}}^*} \right] = \frac{1}{N} \left( \mathbb{E}[\|b_{\hat{F}}\|]^2 - \|\mu\|^2 \right) \tag{18}$$

This is an improvement over $\mathrm{Var}[\hat{m}_{\mathrm{CMC}}]$ because $\mathbb{E}[\|b_{\hat{F}}\|]^2 \leq \mathbb{E}[\|b_{\hat{F}}\|^2]$, as can be seen, for example, by applying Jensen's inequality [13].

We place ourselves in a more general context, corresponding to a non-constant cost $\hat{c}$. More precisely, we assume the following:

- The *Poisson process approximation* (justified in Section 3.14):

$$\|\mu\|^2 \ll \mathbb{E}\left[ w_q(\hat{F}) \|b_{\hat{F}}\|^2 \right] \tag{19}$$

- The cost $\hat{c}$ being a function of the simulated fire:

$$\hat{c} = c(\hat{F}) \tag{20}$$

The cost-to-precision ratio achieved by a proposal distribution $q$ is then as follows:

$$r_{C/P}^{(q)} = \mathbb{E}\left[c(\hat{F}^{(q)})\right] \mathbb{E}\left[\frac{p(b_{\hat{F}})}{q(b_{\hat{F}})}\|b_{\hat{F}}\|^2\right] = \mathbb{E}\left[\frac{q(b_{\hat{F}})}{p(b_{\hat{F}})}c(\hat{F})\right] \mathbb{E}\left[\frac{p(b_{\hat{F}})}{q(b_{\hat{F}})}\|b_{\hat{F}}\|^2\right]$$

$$r_{C/P}^{(q)} = \mathbb{E}\left[w_q(\hat{F})^{-1}c(\hat{F})\right] \mathbb{E}\left[w_q(\hat{F})\|b_{\hat{F}}\|^2\right] \tag{21}$$

For practical purposes, we often need to know the weight function $w_q(\cdot)$ only up to a proportionality constant. For example, we can compute the *efficiency gain* $g_q$ of proposal distribution $q$ as follows:

$$g_q := \frac{r_{C/P}^{(p)}}{r_{C/P}^{(q)}} = \frac{\mathbb{E}\left[c(\hat{F})\right]\mathbb{E}\left[\|b_{\hat{F}}\|^2\right]}{\mathbb{E}\left[w_q(\hat{F})^{-1}c(\hat{F})\right]\mathbb{E}\left[w_q(\hat{F})\|b_{\hat{F}}\|^2\right]} \tag{22}$$

The efficiency gain $g_q$ quantifies how the cost-to-precision ratio improves from the Crude Monte Carlo estimator ($\hat{m}_{\mathrm{CMC}}$) to importance sampling ($\hat{m}_q$). We see that replacing $w$ by $\lambda w$ in Equation (22) leaves the result unchanged, because the constant $\lambda$ gets simplified away in the denominator. The same is true for Equation (21).

*3.5. Computing Sampling Frequency Multipliers*

To better understand the effect of importance sampling with some reweighting function $w := p/q$, we introduce the notion of *frequency multipliers*, which describe how the sampling frequencies of simulated fires change when we use importance sampling.

The *same-cost frequency multipliers* correspond to the following procedure. Assume that we initially planned to simulate $N$ fires with the natural distribution $p$, which means an expected computational cost of $N\mathbb{E}\left[c(\hat{F})\right]$. We then choose to adopt importance sampling with reweighting function $w$ (or, equivalently, with proposal density $q$) and consequently adjust the number of simulated fires to $N'$ to keep the same expected computational cost, i.e., $N'\mathbb{E}\left[c(\hat{F}^{(q)})\right] = N\mathbb{E}\left[c(\hat{F})\right]$. For any potential simulated fire $b$, we can then ask the following question: how much more frequently does $b$ get drawn by importance sampling than by our original sampling design? This number is the *same-cost frequency multiplier* of $b$ and can be computed as follows:

$$\frac{N'q(F)}{Np(F)} = \frac{\mathbb{E}\left[c(\hat{F})\right]}{\mathbb{E}\left[w_q(\hat{F})^{-1}c(\hat{F})\right]}w_q(F)^{-1} \tag{23}$$

Similarly, the *same-precision frequency multiplier* can be conceived. Then, the number of simulated fires is adjusted from $N$ to $N''$ so that the precision remains the same, i.e., $N'\mathbb{E}\left[w_q(\hat{F})\|b_{\hat{F}}\|^2\right]^{-1} = N\mathbb{E}\left[\|b_{\hat{F}}\|^2\right]^{-1}$. Then, the frequency multiplier gets computed as follows:

$$\frac{N'q(F)}{Np(F)} = \frac{\mathbb{E}\left[w_q(\hat{F})\|b_{\hat{F}}\|^2\right]}{\mathbb{E}\left[\|b_{\hat{F}}\|^2\right]}w_q(F)^{-1} \tag{24}$$

**Confusion pitfall:** note that the frequency multipliers are *inversely* proportional to the reweighting function $w$.

*3.6. The Optimal Proposal Distribution*

This section shows that there is an optimal proposal distribution and that it can be described quantitatively. Applying the probabilistic Cauchy–Schwarz inequality (Appendix A.2)

to Equation (21) yields an **expression for the optimal proposal distribution** $q_c^*$ and its cost-to-precision ratio:

$$w_{q_c^*}(F)^2 \propto \frac{c(F)}{\|b_F\|^2}$$

$$q_c^*(F) = \mathbb{E}\left[\frac{\|b_{\hat{F}}\|}{c(\hat{F})^{\frac{1}{2}}}\right]^{-1} \frac{\|b_F\|}{c(F)^{\frac{1}{2}}} p(F) \tag{25}$$

$$r_{C/P}^{(q_c^*)} = \mathbb{E}\left[c(\hat{F})^{\frac{1}{2}}\|b_{\hat{F}}\|\right]^2 \tag{26}$$

We define the **maximum potential efficiency gain** $g^*$ as the efficiency gain (Equation (22)) achieved by the optimal proposal distribution $q_c^*$. By combining Equations (22) and (26), we obtain the following expression for $g^*$:

$$g^* = \frac{\mathbb{E}\left[c(\hat{F})\right]\mathbb{E}\left[\|b_{\hat{F}}\|^2\right]}{\mathbb{E}\left[c(\hat{F})^{\frac{1}{2}}\|b_{\hat{F}}\|\right]^2} \tag{27}$$

*3.7. Application to Burn Probability Maps*

In the case of burn maps, recalling that $\|b_{\hat{F}}\|^2 = a(\hat{F})$, it follows that

$$q_c^*(F) = \mathbb{E}\left[\left(\frac{a(\hat{F})}{c(\hat{F})}\right)^{\frac{1}{2}}\right]^{-1}\left(\frac{a(F)}{c(F)}\right)^{\frac{1}{2}} p(F) \tag{28}$$

In other words, the best proposal distribution **reweights the probability of each fire by the square root of its size-to-cost ratio.** Two special cases of cost functions $c(F)$ stand out:

- When the cost of simulating a fire is proportional to its size, then **importance sampling makes no difference:** the initial distribution $p(F)$ is already the optimal proposal distribution;
- When the cost of simulating a fire is constant, then the optimal proposal distribution reweights the probability of each fire by the square root of its size.

It is also interesting to consider a mixture of these cases. The cost function can be modeled as the sum of a constant term and a term proportional to fire size: $c(F) = c_0(1 + a(F)/A_0)$. For very large fires ($A \gg A_0$), the size-to-cost ratio is very close to 1, such that no reweighting occurs: large fires get sampled according to their natural distribution. For very small fires ($A \approx 0$), the size-to-cost ratio is very small, such that these fires have virtually no chance of being sampled; since they do not contribute much to the mean, the weight is not high enough to compensate this sampling scarcity. This provides a principled justification for the common heuristic of dispensing with simulating small fires: provided that $A_0$ is not too large, that heuristic is very close to the optimal proposal distribution.

To further illustrate this phenomenon, Figure 1 displays the maximum potential efficiency gain (Equation (27)) for an imaginary model in which the cost $c$ is determined by the fire size $A$ as $c \propto (1 + (a(F)/A_0)^\epsilon)$, based on the distribution of fire sizes recorded in the Fire Occurrence Database [14]. When the constant term is high (high $A_0$), a lot of efficiency can be gained by reducing the sampling frequency of small fires. When the cost is a superlinear function of size ($\epsilon > 1$), efficiency can be gained by reducing the sampling frequency of large fires.

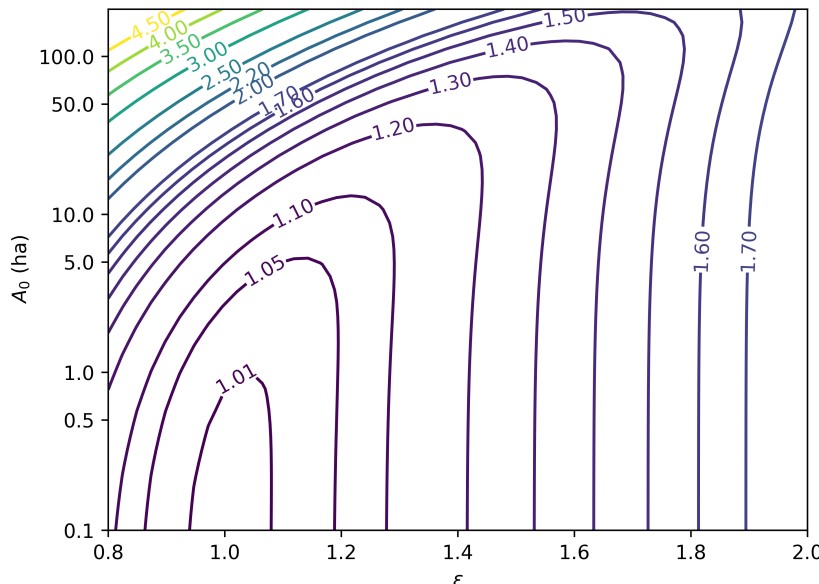

**Figure 1.** Contour plot of the maximum potential efficiency gain from importance sampling for an imaginary model in which the cost $c$ is proportional to a constant term plus the fire size $A$ raised to the exponent $\epsilon$ ($c = c_0(1 + (a(F)/A_0)^\epsilon)$). The underlying distribution of fire sizes is the one recorded in the FPA-FOD [14] in the Conterminous United States from 1992 to 2020. $A_0$ is the natural size constant determined by the constant term. Note that the value of $c_0$ is immaterial.

*3.8. Generalization: Optimal Proposal Distribution for Ignitions*

We now assume that we do not have enough mastery over $p(F)$ to find the optimal proposal distribution $q_c^*$. Instead, we restrict ourselves to seeking a proposal distribution for the ignition $\hat{I}$, such that $q(F, I)$ is of the following form:

$$q(F, I) = p(F|I)q^I(I) \tag{29}$$

This decomposition is consistent with the idea that we are going to change the distribution of ignitions without changing the fire spread algorithm itself. For example, Section 4.4 changes the sampling distribution of ignitions based on fire duration. Under this constraint, the cost-to-precision ratio of the proposal distribution can then be expressed as follows:

$$r_{C/P}^{(q)} = \mathbb{E}\left[\frac{q^I(\hat{I})}{p(\hat{I})}c_I(\hat{I})\right]\mathbb{E}\left[\frac{p(\hat{I})}{q^I(\hat{I})}a_I(\hat{I})\right] \tag{30}$$

in which the functions $a_I(I), c_I(I)$ compute the conditional expectation of the fire size and computational cost:

$$
\begin{aligned}
a_I(I) &:= \mathbb{E}\left[\|b_{\hat{F}}\|^2|\hat{I} = I\right] = \mathbb{E}\left[a(\hat{F})|\hat{I} = I\right] \\
c_I(I) &:= \mathbb{E}\left[c(\hat{F})|\hat{I} = I\right]
\end{aligned} \tag{31}
$$

The proof to Equation (25) can then be generalized to show that the optimal proposal distribution $q_c^{I*}$ is given by the following:

$$q_c^{I*}(\hat{I}) \propto \left(\frac{a_I(\hat{I})}{c_I(\hat{I})}\right)^{\frac{1}{2}} p(\hat{I}) \tag{32}$$

It can then be seen that $q_c^{I*}$ yields the following density and cost-to-precision ratio:

$$q_c^{I*}(F, I) = \mathbb{E}\left[\left(\frac{a_I(\hat{I})}{c_I(\hat{I})}\right)^{\frac{1}{2}}\right]^{-1} \left(\frac{a_I(I)}{c_I(I)}\right)^{\frac{1}{2}} p(F, I)$$

$$r_{C/P}^{(q_c^{I*})} = \mathbb{E}\left[\left(c_I(\hat{I}) a_I(\hat{I})\right)^{\frac{1}{2}}\right]^2 \tag{33}$$

Note that any random variable $\hat{I}$ can be chosen for these results to apply. Here, we have described $\hat{I}$ as the "ignition" or "initial conditions", but it can be any predictor of the fire size and simulation cost. Two extreme cases of $\hat{I}$ are noteworthy:

- If $b_{\hat{F}}$ is fully determined by $\hat{I}$ (deterministic simulation), then we recover $q_c^{I*} = q_c^*$.
- If $\hat{I}$ is uninformative (independent) with respect to $b_{\hat{F}}$, then we obtain a no-progress optimum of $q_c^{I*} = p$.

*3.9. Generalization: Impact-Weighted $L_2$ Distance*

An interesting generalization consists of using a measure other than area for computing $L_2$ distances by introducing a density function $u(s)$ that quantifies the "value" or "concern" per unit area:

$$\|z\|_u^2 := \int_s |z(s)|^2 u(s) ds \tag{34}$$

Then, all the results of the previous section apply (with $ds$ being replaced by $u(s)ds$ in integrals), except that the interpretation changes: quantities like $\|b_{\hat{F}}\|_u^2$, $a(\hat{F})$ and $a_I(\cdot)$ no longer measure fire size but burned value. In particular, when using the optimal proposal distribution for importance sampling, each fire sees its sampling probability **reweighted by the square root of its burned value/simulation cost ratio.**

Weighting by value instead of area can dramatically change the optimal proposal distribution. It can boost the probability of sampling small fires near valuable assets while depressing the probability of large fires in low-concern areas. From this perspective, the "area of interest" takes on a very precise meaning: the set of locations $s$ where $u(s) > 0$.

The choice of value measure $u(s)ds$ depends on how the burn probability map will be used: where do we want precision? For example, the burn probability maps might be used to plot the risk to properties and infrastructure and also be aggregated to estimate GHG and smoke emissions. To that end, a good value function could place strong measure on properties and infrastructure and mix that with small measure on fuel loads (the latter do not need as high a measure because emissions are destined to be aggregated, which already reduces variance).

*3.10. Generalization: Non-Geographic Maps and Non-Binary Outcomes*

Another way to understand Section 3.9 is to pretend that the variable $s \in \mathcal{A}$ ranges not over geographic space but over "assets space". Attentive readers may have noticed that nothing in the above analysis is specific to the spatial nature of the set $\mathcal{A}$. We can conceive of other types of burn probability maps, for example, $\mathcal{A}$ may be as follows:

1. A space–time region, like $\mathcal{A} :=$ California $\times$ [2024, 2034) or $\mathcal{A} :=$ Alberta $\times$ Months, in which Months := {January, February, . . . , December};
2. A combination of geographic space and fire severity buckets, like $\mathcal{A} :=$ Australia $\times$ {Severity Class 1, Severity Class 2, . . . };
3. A combination of geographic space and paired scenario outcomes, like the following:

$\mathcal{A} :=$ area of interest

$\times$ {"burned without fuel treatment", "did not burn without fuel treatment"}

$\times$ {"burned with fuel treatment", "did not burn with fuel treatment"}

The above paragraph generalizes $\mathcal{A}$, that is, the *domain* of function $s \mapsto b_{\hat{F}}(s)$; we can also generalize its *codomain*, that is, the simulation outcomes. In particular, there are the following:

- $b_{\hat{F}}(s)$ **need not be binary:** for example, $b_{\hat{F}}(s)$ might be a density of greenhouse-gas emissions;
- $b_{\hat{F}}(s)$ **need not be about fire.**

What matters for the analysis of this article to apply is that the $b_{\hat{F}}$ outcomes be aggregated by averaging and for the *Poisson process approximation* to be valid (Section 3.14). Note that when the outcomes are not binary, $\|b_F\|^2$ can no longer be interpreted as the impacted area, i.e., Equation (11) ($a(F) = \|b_F\|^2$) no longer holds.

### 3.11. Variant: Sampling from a Poisson Point Process

The above analysis assumes that the number of fires is fixed and that the fires are sampled i.i.d. A variant of this sampling scheme is to sample from a Poisson Point Process (PPP) [15] instead: this is equivalent to saying that the number of fires is no longer fixed but follows a Poisson distribution. The PPP can be appealing because it is algorithmically convenient (subregions of space–time can be sampled independently) and yields mathematical simplifications, as we show below.

In the context of a PPP, a random number $\hat{N}_q \sim \mathrm{Poisson}(\lambda = N)$ of fires is sampled, and the Likelihood Ratio estimator, which we now denote as $\ddot{m}_q$, is virtually unchanged:

$$\ddot{m}_q := \frac{1}{N} \sum_{k=1}^{\hat{N}_q} w_q(\hat{F}_k^{(q)}) b_{\hat{F}_k^{(q)}} \tag{35}$$

By writing $\mathbb{E}[\ddot{m}_q] = \mathbb{E}[\mathbb{E}[\ddot{m}_q|\hat{N}_q]]$, it is easily verified that the mean of this estimator is correct: $\mathbb{E}[\ddot{m}_q] = \mu$.

An interesting aspect of the PPP is that the variance approximation of Equation (19) becomes exact (this is why we named it the *Poisson process approximation*), such that we have the following:

$$\mathrm{Var}[\ddot{m}_q] = \frac{1}{N} \mathbb{E}\left[w_q(\hat{F}) \|b_{\hat{F}}\|^2\right] \tag{36}$$

It follows that the results of the previous section about the optimal proposal distribution and cost-to-precision ratio remain unchanged.

### 3.12. Practical Implementation

The previous sections described the optimal proposal distribution, which can be achieved with perfect knowledge of the impact and cost functions $a_I(\cdot)$ and $c_I(\cdot)$. In practice, these functions will be unknown so that Equation (32) is not applicable. Instead, we suggest using a calibration approach to estimate an approximately optimal proposal distribution through the following steps:

1. Run simulations to draw a sample of ignition and simulated fire $(\hat{I}, \hat{F})$, yielding an empirical distribution, which we represent by the random variables $(\tilde{I}, \tilde{F})$.
2. Apply Equation (21) to $\tilde{F}$ instead (using $\tilde{F}$ of $\hat{F}$ makes it possible to compute expressions like $\mathbb{E}[h(\tilde{F})]$ from the empirical sample, without full knowledge of the underlying distribution $p(\cdot)$ of $\hat{F}$ to estimate the potential gain in the cost-to-precision ratio. If that gain is not significant, abandon the importance sampling.
3. Design a parametric family of weight functions $w(I; \theta)$.
4. Find the optimal parameter vector $\theta^*$ by solving the following optimization problem, inspired by Equation (21):

$$\theta^* := \mathrm{argmin}\left(\theta \mapsto \mathbb{E}\left[w(\tilde{I}; \theta)^{-1} c(\tilde{F})\right] \mathbb{E}\left[w(\tilde{I}; \theta) a(\tilde{F})\right]\right) \tag{37}$$

If $\theta^*$ does not yield a significant improvement compared to $w(I) = 1$, try a new family of weight functions or abandon the importance sampling.

5.  Adopt $q^I(I) \propto w(I, \theta^*)^{-1} p(I)$ as the proposal distribution for the importance sampling.

Care should be taken to appropriately constrain the $(w(\cdot; \theta))_\theta$ parametric family to avoid any problem of identifiability, since the objective function in Equation (37) will yield exactly the same score for a reweighting function $w$ and a scaled variant $\lambda w$; therefore, the family of reweighting functions must not be stable by rescaling. It might also be sensible to add a regularization term to the objective function.

In addition to using a parametric family, it can be sensible to use a curve-fitting approach: one can compute the weights that are optimal for the (discrete) empirical distribution and then use some interpolation algorithm to derive a continuous weight function, like a supervised learning algorithm or nearest-neighbor method [16].

### 3.13. Analysis of Geographic Inequalities in Precision

The above analysis describes how to optimize the global precision (in the sense of the $L_2$ norm); however, doing so might alter the relative geographic distribution of local precision across the map. In particular, we can expect that this optimization will favor absolute precision over relative precision, spending the computational cost mostly in areas of high burn probability. Section 4.3 provides an example that illustrates this phenomenon.

### 3.14. Justifying the Poisson Process Approximation

The proof to Equation (25) relies on the Poisson process approximation (Equation (19)), which we now justify. It follows from Equation (18) that $\mathbb{E}\left[\|b_{\hat{F}}\|\right]^2$ is the smallest possible value for $\mathbb{E}\left[w_q(\hat{F})\|b_{\hat{F}}\|^2\right]$. Therefore, it is enough to show that $\mathbb{E}\left[\|b_{\hat{F}}\|\right]^2 \gg \|\mu\|^2$. We suggest that this generally holds by considering the case where $\mu$ is spatially uniform. Denoting $|\mathcal{A}|$ the total area, we then have $\mu(s) = |\mathcal{A}|^{-1} \mathbb{E}\left[a(\hat{F})\right]$, and $\|\mu\|^2 \ll \mathbb{E}\left[a(\hat{F})^{\frac{1}{2}}\right]^2$ is equivalent to the following condition on the *burned fraction* $a(\hat{F})/|\mathcal{A}|$:

$$\mathbb{E}\left[\frac{a(\hat{F})}{|\mathcal{A}|}\right] \ll \mathbb{E}\left[\left(\frac{a(\hat{F})}{|\mathcal{A}|}\right)^{\frac{1}{2}}\right] \tag{38}$$

which is the case in the typical situation where the burned fraction is almost certainly much smaller than 1. The general phenomenon here is that $\|\mu\|^2$ becomes an insignificant component of variance as the AOI is made much larger than the fires; this corresponds to the Binomial-to-Poisson approximation we showed in a previous paper [17].

Another perspective to shed light on this phenomenon is that making the AOI very large means that the sampling procedure behaves like a Poisson Point Process (Section 3.11) in any minority subregion of the AOI; then, Equation (36) holds approximately. This is the same phenomenon as the Binomial-to-Poisson approximation: the number of points drawn in a small subregion follows a Binomial distribution with a small probability parameter, which is well approximated by a Poisson distribution.

### 3.15. Importance Sampling Versus Stratified or Neglected Ignitions

Stratification [18] is another strategy for reducing the variance of Monte Carlo simulations. In the case of burn probability estimation, stratification might consist of partitioning the set of possible ignitions into several subsets (for example, low risk and high risk) and simulating more fire seasons in some subsets than others. The burn probability maps from each subset are then combined by a weighted sum, the weights being inversely proportional to the number of fire seasons. A similar strategy consists of dividing the space of ignitions between low-risk and high-risk subsets, and simulating only the high-risk subset, thus neglecting the other.

Observe that both strategies are highly similar to a special case of importance sampling, in which the weight function $w_q(\cdot)$ is piecewise constant. Therefore, importance sampling is useful for reasoning about these other strategies and provides an upper bound to their potential efficiency gains.

## 4. Results

We illustrate the above methods by applying them to an existing dataset of results from a wildfire Monte Carlo simulation. The next sections implement various forms of importance sampling, estimate the efficiency gains, and discuss the tradeoffs. Due to limited access to computed resources, we did not run new simulations based on the reweighted sampling scheme; instead, we used simulation results that were drawn without importance sampling and quantified the efficiency gains that would have been obtained by implementing importance sampling.

### 4.1. Simulation Design and Results

The dataset of simulated fires studied here was produced as part of the 3rd version of the FireFactor project [4], which provides simulation-based predictions of wildfire risk over the Contiguous United States (CONUS). The rest of this paragraph provides some details to make the simulation setup more tangible; however, note that these characteristics hardly have any bearing on our application of importance sampling and are only provided for context (and without justification, as that would far exceed the scope of the present paper). Fires were simulated with the ELMFIRE fire behavior model [19], using weather inputs derived from the Real-Time Mesoscale Analysis (RTMA) dataset [20], combined with Oregon State PRISM (Parameter-elevation Regressions on Independent Slopes Model) [21], from which fuel moisture weather variables were derived using standard NFDRS calculations [22]; fuels and topography inputs were derived from the LANDFIRE dataset [23], edited to account for more recent disturbances. A total of 800 repetitions of the 13 years from 2011 to 2023 were simulated with historical weather, with the assumption that this time period is representative enough of the short-term distribution of future fire occurrence. The distribution of simulated fires was calibrated using the PiRPLO statistical framework [24] by fitting parametric models of ignition density and fire duration distribution to the empirical distribution of ignition coordinates and fire sizes from the Fire Occurrence Database (FOD) [14] in the 2011–2020 time period. According to the FOD, 26 Mha were burned in this time period by 759,000 recorded fires, including 11,900 fires larger than 100 ha.

We do not try to discuss the validity and implementation details of these simulations here—that would require its own publication—and we henceforth focus solely on the aspect relevant to our study of importance sampling.

The ignition locations, times, and fire durations were randomly drawn as a Poisson Point Process [15], following a density function that was influenced by predictor variables such as fuel type, weather, distance to built areas, and a spatial density of ignitions derived from historical data. Compared to previous versions of the FireFactor project, a salient feature of this version is a heavy tail of fire durations, which is an attempt to represent the occurrence of very large fires more faithfully. As already noted, we did not use importance sampling in these simulations.

This simulation procedure produced a dataset of approximately 31.7 million simulated fires. Because this amount of data was impractical for the experiments described below, we thinned this dataset into a random sub-sample of approximately 2.7 million fires: for each fire, we independently drew a random binary variable to determine whether it would be selected in the sub-sample. To improve precision, the larger fires had a higher probability of being selected in the sub-sample (astute readers will have noticed that this subsampling strategy is yet another form of importance sampling). Based on a rough examination of a

fire size histogram and some judgment-based trial and error, the selection probability $p_s$ was chosen to be the following function of the fire size $A$:

$$p_s(A) := \min\left(1, \left(\frac{2000 \text{ ac} + A}{2000 \text{ ac} + 30000 \text{ ac}}\right)^{1.2}\right) \tag{39}$$

Thus, the selection probability was almost uniform when $A \ll 2000$ ac (809.3 ha), approximately proportional to $A^{1.2}$ when $A \gg 2000$ ac, and saturating to 1 when $A \geq 30,000$ ac (12,140.6 ha). This selection probability function is largely arbitrary, so we do not pretend to provide a theoretical justification for it; fortunately, it can be justified empirically by estimating the sensitivity of downstream results to the sampling randomness, which we did by bootstrap-resampling the sub-sample [25].

Denote $(F_i)_i$ as the original sample of 31.7 million fires and $(F'_k)_k$ as the thinned sub-sample. Then, for any function $u(F)$, we can approximate the sum of the $u(F_i)$ as follows:

$$\sum_i u(F_i) \overset{\mathbb{E}}{\approx} \sum_k s_k u(F'_k) \tag{40}$$

in which the symbol $\overset{\mathbb{E}}{\approx}$ denotes equality in expectation, and $s_k$ is the appropriate *subsampling weight* derived from the selection probability:

$$s_k := \frac{1}{p_s(A'_k)} \tag{41}$$

### 4.2. Maximum Potential Efficiency Gain

By applying Equation (27) to the dataset described in Section 4.1, we obtained $g^* = 1.56$ (standard error $\sigma = 0.005$). The standard error $\sigma$ was estimated by bootstrap-resampling [25] using 100 bootstrap replications.

The above result is an upper bound on the efficiency gain that can be obtained by importance sampling; however, it does not tell us what reweighting function to use in order to approach that upper bound. The next sections explore various reweighting strategies.

### 4.3. Reweighting Pyromes Uniformly

We start with a reweighting function that is constant within each pyrome. The approach is therefore similar to stratifying (Section 3.15) by pyrome. Pyromes [26] are a spatial partition of the Conterminous United States into 128 areas of relatively homogeneous fire occurrence characteristics. This reweighting strategy means that instead of simulating the same number of fire seasons in all pyromes, we simulate more fire seasons in some pyromes than others; however, the relative frequencies of simulated fires ignited within a given pyrome remain the same.

By applying Equation (33) with variable $I$ defined as the pyrome of ignition, we know the importance weight for a pyrome is proportional to $\sqrt{c(I)/a(I)}$, the square root of the expected cost divided by the expected burned area in that pyrome. Thus, we estimated $a(I)$ and $c(I)$ for each pyrome $I$, yielding the optimal weighting function, up to some proportionality constant. Using Equation (22), we computed the efficiency gain $g_q$, yielding a value of $g_q = 1.09$.

We then assessed the geographic inequality in precision implied by this reweighting scheme (Section 3.13). To do so, we computed the same-cost frequency multipliers (Equation (23)), telling us how the number of simulated seasons in each pyrome has increased or decreased. All else being equal, precision is proportional to the number of simulated years; therefore, the gain in precision within each pyrome is equal to this multiplier. Table A1 shows the multipliers for each pyrome, along with some other descriptive statistics; these numbers are plotted in Figure 2. We see that most pyromes have a multiplier between 0.7 and 1.3, with a few as low as 0.4 and none higher than 2. A total of 65 out of

128 pyromes have a multiplier between 0.9 and 1.1, which means that they are little affected by the importance sampling.

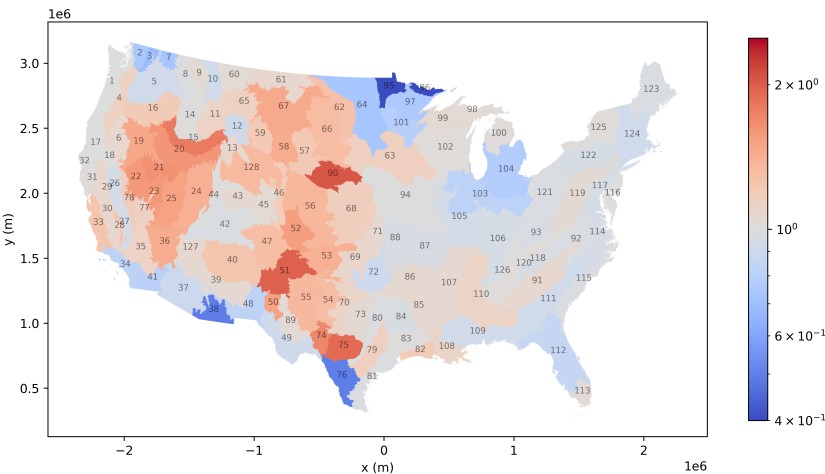

**Figure 2.** Same-cost frequency multipliers for the optimal importance sampling scheme that reweights each pyrome uniformly. Each pyrome is colored by its frequency multiplier (note in particular that the color does not represent the prevalence of fire in each pyrome). The color bar is logarithmic and saturates to blue at 0.4 and to red at 2.5. More details in Table A1.

In our opinion from looking at these results, this particular reweighting scheme is more trouble than it is worth: the efficiency gain is rather small, and the reduced precision in some pyromes is concerning.

### 4.4. Duration-Based Parametric Reweighting in Pyrome 33

As another example, we restrict our attention to only one pyrome and use a parametric weight function (as suggested in Section 3.12) based on the simulated fire duration. The assumption here is that the simulated fire duration is randomly drawn at the same time as the ignition location (before the fire is simulated) and that we can oversample or undersample fires based on their duration.

The pyrome under consideration is Pyrome 33 (*South Central California Foothills and Coastal Mountains*) (the choice of this pyrome was largely arbitrary: we noticed that $g^*$ was large enough to make further investigation worthwhile, and this geographic area was of some interest to us in other research projects; we stress that the specific fire ecology of Pyrome 33 is of no relevance to the present material). This pyrome is between the Pacific Coast and the Central Valley of California, roughly stretching between the cities of San Francisco to the North and Santa Maria to the South. Almost all of its area is comprised in the *Mediterranean California* CEC ecoregion [27]. For this pyrome, we computed (Equation (27)) a maximum potential efficiency gain of $g^* = 1.38$.

We use a parametric reweighting function of the following form:

$$\ln w_1(t; \alpha) = \text{const} + \alpha \exp\left(-\frac{t}{150\,\text{h}}\right) \tag{42}$$

in which $\alpha$ is a parameter controlling how short fires get oversampled ($\alpha < 0$) or undersampled ($\alpha > 0$) by importance sampling.

To estimate the optimal value $\alpha^*$, we used the `scipy.optimize.minimize` function of the SciPy library [28] to solve the following optimization problem, corresponding to Equation (37):

$$\alpha^* = \text{argmin}\left(\alpha \mapsto \left(\sum_k s_k w_1(t_k; \alpha)^{-1} c(F_k)\right)\left(\sum_k s_k w_1(t_k; \alpha) a(F_k)\right)\right) \tag{43}$$

in which $k$ indexes the simulated fires of the calibration dataset, $t_k$ is the duration of the $k$-th simulated fire, and $s_k$ is the subsampling weight (see Equation (41)).

The best-fitting parameter value is $\alpha^* = -1.95$, achieving an efficiency gain of 1.25 (as computed by Equation (22)). Shorter fires obtain a smaller weight ($\alpha^* < 0$), which means that importance sampling will improve efficiency by over-representing them. The curve in Figure 3 shows how importance sampling alters the sampling frequency of fires under this reweighting function by displaying the same-cost frequency multipliers (Equation (23)) corresponding to $w(\cdot, \alpha^*)$, along with a point cloud of the frequency multipliers yielded by the best possible reweighting scheme (given by Equation (25)).

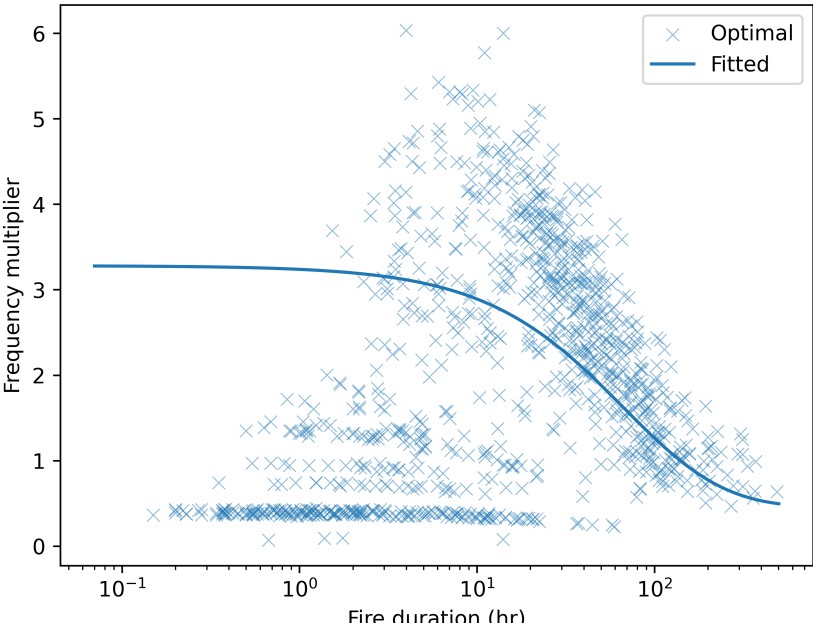

**Figure 3.** Same-cost frequency multipliers for the reweighting function fitted to Pyrome 33 (solid curve), along with the best possible reweighting function (point cloud) of the calibration sample. $(x, y)$ means the following: for the same compute time, importance sampling multiplies by $y$ the sampling frequency of a fire of duration $x$.

A side effect of the reweighted function fitted above is that it increases the sampling frequency of very short fires. There is cause to be unsatisfied with this, in particular because simulating a very large number of tiny fires can make the results unwieldy in ways not captured by our cost function. To address this limitation, we now fit a more refined reweighting function, expressed as follows:

$$\ln w_2(t; \alpha_1, \tau_1, \alpha_2, \tau_2) = \text{const} + \alpha_1 \exp\left(-\frac{t}{\tau_1}\right) + \alpha_2 \exp\left(-\frac{t}{\tau_2}\right) \tag{44}$$

The above function is more flexible, having four parameters instead of one; in particular, the duration lengthscales are no longer fixed, thanks to the $\tau_i$ parameters. We fitted this function with initial parameters ($\alpha_1 = 0, \tau_1 = 5\,\text{h}, \alpha_2 = 0, \tau_1 = 150\,\text{h}$): in practice, this means that ($\alpha_1, \tau_1$) influence short fires, while ($\alpha_2, \tau_2$) influence long fires. The best-fit values were ($\alpha_1^* = 2.53, \tau_1^* = 3.63\,\text{h}, \alpha_2^* = -2.15, \tau_2^* = 81.2\,\text{h}$), achieving an efficiency gain of 1.27 (confirmed with K-fold cross-validation). Note that $\alpha_1^* > -\alpha_2 > 0$, thus reducing the sampling frequency of very short fires. Figure 4 displays the fitted frequency multipliers: in summary, importance sampling in this case will increase the sampling frequency of medium-duration fires at the expense of low-duration and high-duration fires.

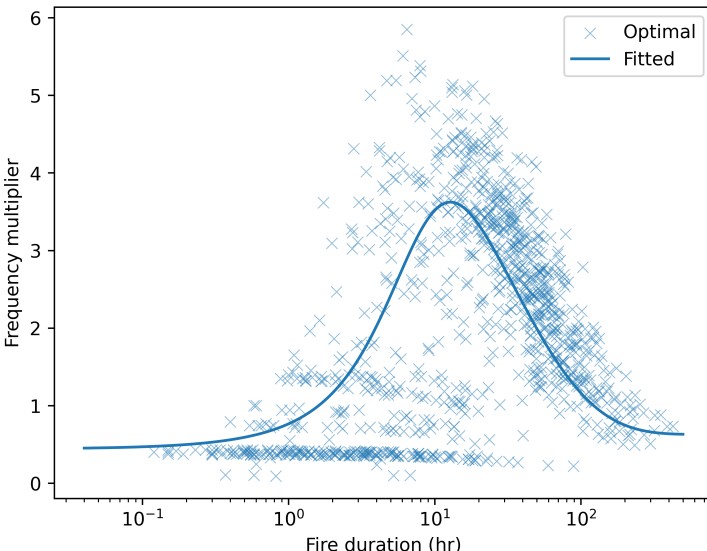

**Figure 4.** Same-cost frequency multipliers for the $w_2$ reweighting function fitted to Pyrome 33 (solid curve), along with the best possible reweighting function (point cloud) of the calibration sample.

### 4.5. Fitting a Machine Learning Model to Empirical Optimal Weights

We now illustrate the last suggestion of Section 3.12: fitting supervised learning to the optimal weights derived from the empirical sample. Using the dataset of 2.7 million simulated fires described in Section 4.1, we trained a gradient-boosted regression tree algorithm [16] to predict the optimal weight $w^*$ (computed from the empirical sample using Equation (25)). More precisely, we used the `HistGradientBoostingRegressor` class of the Scikit-Learn library [29], which is inspired by LightGBM [30]. We used $\ln w^*$ as the target variable, and we weighted the samples by fire size. As predictor variables, we used the fire duration, pyrome number, ignition-point fuel model number, ignition-time ERC percentile, some fire behavior metrics (rate of spread, reaction intensity in $kW/m^2$, and fire-line intensity in $kW/m$) derived from the fuel model in reference weather conditions (no-slope heading fire with a 10 mi/h mid-flame wind speed; 1 h/10 h/100 h dead fuel moistures of 3%, 4%, and 5%; and live herbaceous/woody fuel moistures of 40%/80%) using the Rothermel model [31], and spatial coordinates of ignition point $(x, y)$.

We fitted two models, differing only in which subset of predictor variables was used:

1. Model A: used all the predictor variables listed above.
2. Model B: used all the predictor variables except for the spatial coordinates $(x, y)$.

The models were tested by a method similar to K-fold cross-validation: training on a random subset of 70% of the fires, computing the efficiency gain on the remaining 30%, repeating this procedure 30 times, and computing the mean and standard deviation of the efficiency gain. The test performance of the model is summarized in Table 2. We see that Model A achieves a much higher efficiency gain than Model B, revealing that it learns fine-grained spatial features from the spatial coordinates of ignition. While its superior efficiency gain is appealing, we are wary of using Model A because of the subtle potential side effects of having estimator precision that varies at fine spatial scales.

**Table 2.** Performance metrics of gradient-boosted tree models fitted to the example dataset. See text for details.

| Model | Efficiency Gain |
| --- | --- |
| Model A | 1.25 ($\sigma = 0.00380$) |
| Model B | 1.15 ($\sigma = 0.00085$) |

## 5. Discussion

Let us review the main contributions of this paper and discuss their implications. Recall that importance sampling is primarily concerned with computing efficiency: importance sampling increases the convergence speed of Monte Carlo simulations toward the target distribution. Computing efficiency is a major concern in wildfire simulations, influencing critical design tradeoffs in simulation software [2]. This justifies an interest in importance sampling for fire simulations. Furthermore, it is well known that most wildfire impact is driven by a small minority of wildfires [32]; this suggests that importance sampling has the potential to be valuable, since it consists of putting more emphasis on the high-impact subregions of the distribution of simulated fires.

Our analysis starts from the classical generic result of the Monte Carlo literature about importance sampling for a scalar estimator, which shows that the variance-minimizing proposal distribution reweights the probability of each sample by its absolute value (see Equation (18)). We have made this result applicable to the estimation of burn probability maps through a series of generalizations:

1.  Estimating a map rather than a scalar (Section 3.2) and using the $L_2$ distance as the metric of deviation underlying variance (Section 3.3);
2.  Accounting for the variability in computational costs across simulated fires and adopting cost-to-precision as the performance metric (Equation (7)) instead of simply variance;
3.  Deriving the optimal cost-to-precision and the corresponding proposal distribution (Section 3.4) and relying on the *Poisson process approximation* (Section 3.14) that each fire affects only a small fraction of the area of interest;
4.  Generalizing to proposal distributions constrained to an upstream variable (typically the ignition), see Section 3.8;
5.  Generalizing to impact-weighted $L_2$ distance (Section 3.9);
6.  Generalizing from burn probability estimation to other types of maps (Section 3.10).

We illustrated these methods in Section 4 with various case studies. Here is a summary of the **key findings:**

1.  Section 3.3 observes that the relevant metric for quantifying the efficiency of a Monte Carlo simulation is the cost-to-precision ratio $r_{C/P}$, that is, the product of variance to expected computational cost (Equation (7)).
2.  Without importance sampling, the variance of the burn probability estimator is simply the expected burned area divided by the number of simulated fires (Equations (13) and (6)).
3.  The efficiency characteristics of the Monte Carlo sampling, and in particular the proposal distribution which maximizes convergence speed, are fully determined by the joint distribution of the fire size and computational cost (Equation (21)). Beyond these two variables, the diversity of the shapes and behaviors of the simulated fires is irrelevant.
4.  The optimal proposal distribution (lowest cost-to-precision ratio) reweights the natural probability of each fire by a factor proportional to the square root of the ratio of burned area to the computational cost (Equation (25)). When only the distribution of ignitions can be reweighted, the burned area and computational cost must be replaced by their expectation conditional on the ignition (Equation (33)).
5.  In particular, when the computational cost is strictly proportional to the burned area, importance sampling can achieve no progress: the natural distribution is already the optimal proposal distribution. This linearity assumption is most dubious at the "small fires" tail of the natural distribution.
6.  Finding a good proposal distribution is not trivial, as it requires predictive power. Section 3.12 suggests a machine learning approach based on calibration runs.
7.  However, the best achievable cost-to-precision ratio is easily estimated based on a calibration sample of fire sizes and computational costs, using Equation (26). This allows for quickly assessing whether importance sampling is worth pursuing.

8. Stratified or neglected ignitions can be seen as approximations to special cases of importance sampling, in which the reweighting is piecewise constant (Section 3.15).

Perhaps the most questionable aspect of these methods is the use of $L_2$ as the metric of deviation. While this choice is mathematically convenient, and has various theoretical justifications, it may create geographic disparities in precision and excessively favor absolute precision over relative precision (as noted in Section 3.13).

When the computational cost of a simulated fire is asymptotically a superlinear function of its size, then importance sampling will reduce the sampling frequency of large fires (Section 3.7). This is unintuitive and can be problematic: the largest fires are typically those we are most interested in. Reducing the sampling frequency of large fires may improve precision in the sense of $L_2$, but it might at the same time conflict with other precision objectives. This issue can be mitigated in several ways:

1. By weighting the $L_2$ metric by value (Section 3.9), e.g., by giving more importance to Highly Valued Resources and Assets;
2. By including non-geographic outcomes like burn severity (Section 3.10) and weighting them unevenly;
3. By constraining the family of reweighting functions under consideration to be uniform in the tail of large fires.

Consequently, the main recommendation of this article is to consider using some form of importance sampling, not necessarily the one which is optimal in the sense of $L_2$ convergence. We find the approach most promising for reducing the sampling frequency of many small inconsequential fires in situations where the simulation pipeline makes them disproportionately costly.

To make such approaches practical, we recommend that wildland fire simulation software enable users to provide weights alongside ignitions, with these weights being taken into account for aggregating the simulation results into outputs. It should be immaterial whether such weights originate from importance sampling or similar techniques, like stratified sampling.

## 6. Conclusions

This article has studied the application of importance sampling to Monte Carlo wildfire simulations. Importance sampling draws simulated fires from a proposal distribution that is different from the target distribution that models fire occurrence. This technique can accelerate the convergence of Monte Carlo simulations to their asymptotic burn probability map; **importance sampling does not introduce bias and therefore cannot improve or degrade the realism of that asymptotic limit.** Simulating an infinite number of fires would yield the same burn probability map with or without importance sampling.

One difficulty in applying importance sampling to wildfire simulations is that the estimand (namely, the burn probability map) is a map rather than a scalar. It is not trivial to decide how to quantify convergence speed in this multidimensional setting: our approach is to use the $L_2$ distance as the metric of deviation, which is analytically convenient. This has enabled us to quantify efficiency through quantities like the cost-to-precision ratio and to provide various optimality bounds and search methods for finding a well-performing proposal distribution.

We do not necessarily recommend applying the methods and formula of this paper to the letter. In our opinion, the most useful contributions of this work are to provide a relevant framing of the computational efficiency problem, to raise awareness of importance sampling in general to the fire modeling community, and to showcase some calculation techniques associated with it. The particular flavor of importance sampling studied here, which uses $L_2$ distance as the metric of convergence, is more questionable and should be assessed on a case-by-case basis (see Section 5). Ultimately, the optimal proposal distribution depends on the intended downstream use of the simulation results.

**Supplementary Materials:** The following supporting information can be downloaded at https://www.mdpi.com/article/10.3390/fire7120455/s1, which includes source code files: importance_sampling.py and requirements.txt.

**Author Contributions:** Conceptualization, Methodology, and Formal analysis: V.W.; Investigation and Software Implementation: V.W.; Validation: V.W.; Writing: V.W.; Funding acquisition, Project administration, and Supervision: D.S.; Visualization: V.W. All authors have read and agreed to the published version of the manuscript.

**Funding:** This research was supported financially by the First Street Foundation as part of developing the latest version of the FireFactor product.

**Data Availability Statement:** The Python source code for the figures in this study are included in the Supplementary Materials (file importance_sampling.py). The Fire Occurrence Database data are publicly available at https://doi.org/10.2737/RDS-2013-0009.6. The dataset of simulation results is available upon request. The dataset of simulation results used in this study is available upon request. At this time, we have opted not to publish the dataset as it represents an unfinished product and may not be suitable for purposes beyond illustrating the findings of this study. However, we are more than willing to share and curate the dataset for any researchers who may find it useful for their work. Further inquiries can be directed to the corresponding author.

**Acknowledgments:** The authors thank Teal Richards-Dimitrie for supporting this work.

**Conflicts of Interest:** The authors declare no conflicts of interest.

## Abbreviations

The following abbreviations are used in this manuscript:

| | |
|---|---|
| AOI | Area Of Interest |
| CMC | Crude Monte Carlo |
| PPP | Poisson Point Process |

## Appendix A. Mathematical Background

*Appendix A.1. Probability Basics*

**Notation:** We write $\mathbb{E}[Y]$ for the expected value of random variable $Y$ and $\mathrm{Var}[Y]$ for its variance. $Y \sim Y'$ means that $Y$ and $Y'$ follow the same probability distribution; if $p(y)$ is a probability density function, $Y \sim p$ means that $Y$ follows the corresponding probability distribution. $Y \overset{\mathbb{E}}{\approx} Y'$ means that $Y$ and $Y'$ have the same expected value, i.e., $\mathbb{E}[Y] = \mathbb{E}[Y]'$.

If $Y$ is a random variable with probability density $p(y)$, and $u(y)$ is a function, then the expected value $\mathbb{E}[u(Y)]$ can be written as an integral:

$$\mathbb{E}[u(Y)] = \int_y dy\, p(y) u(y) \tag{A1}$$

*Appendix A.2. The Probabilistic Cauchy–Schwarz Inequality*

The probabilistic Cauchy–Schwarz inequality is the following theorem. For any two real-valued random variables $U$ and $V$, we have the following:

$$\mathbb{E}[UV] \leq \mathbb{E}\left[U^2\right]^{\frac{1}{2}} \mathbb{E}\left[V^2\right]^{\frac{1}{2}} \tag{A2}$$

This inequality is an equality if and only if $U$ and $V$ are proportional to one another with a non-negative coefficient, i.e., if $U = \lambda V$ or $V = \lambda U$ for some $\lambda \geq 0$ (more rigorously, if there exists $\lambda \geq 0$ such that either $\mathbb{P}[U = \lambda V] = 1$ or $\mathbb{P}[V = \lambda U] = 1$).

## Appendix B. Detailed Results of the Per-Pyrome Reweighting Scheme

**Table A1.** Fire statistics and same-cost frequency multipliers for the optimal importance sampling scheme that reweights each pyrome uniformly. The rightmost columns are the expected values per simulated year. If a given pyrome has a multiplier of 1.3, it means that for the same amount of total compute time, the fires will be sampled 1.3 times more frequently from this pyrome under importance sampling than originally. In other words, if the original sampling procedure simulated 10,000 fire seasons in that pyrome, the importance sampling now samples 13,000 fire seasons.

| Pyrome | Frequency Multiplier | Model-Predicted Expected Total | | |
|---|---|---|---|---|
| ID | (Same Cost) | Burned Area (ha/yr) | Runtime (s/yr) | Fires (yr$^{-1}$) |
| 1 | 1.00 | 4191.2 | 21.97 | 50.7 |
| 2 | 0.74 | 1509.5 | 14.17 | 19.6 |
| 3 | 0.68 | 16,112.7 | 181.65 | 26.1 |
| 4 | 1.14 | 6427.3 | 25.61 | 51.9 |
| 5 | 0.92 | 58,477.0 | 359.71 | 98.5 |
| 6 | 1.07 | 19,684.2 | 88.78 | 19.0 |
| 7 | 0.72 | 32,253.7 | 323.36 | 30.9 |
| 8 | 0.99 | 13,646.9 | 73.00 | 22.6 |
| 9 | 1.04 | 20,200.9 | 96.46 | 19.9 |
| 10 | 0.92 | 18,636.2 | 115.63 | 23.7 |
| 11 | 1.13 | 28,773.0 | 118.23 | 26.1 |
| 12 | 0.91 | 5986.2 | 38.03 | 9.6 |
| 13 | 1.07 | 6355.7 | 29.07 | 8.6 |
| 14 | 1.01 | 93,709.1 | 477.19 | 107.5 |
| 15 | 1.79 | 77,990.2 | 126.73 | 86.8 |
| 16 | 1.20 | 83,122.4 | 300.23 | 73.4 |
| 17 | 0.99 | 42,864.7 | 226.30 | 38.0 |
| 18 | 1.02 | 7899.4 | 39.51 | 3.6 |
| 19 | 1.47 | 29,049.8 | 69.55 | 28.0 |
| 20 | 1.61 | 102,857.4 | 206.72 | 70.5 |
| 21 | 1.64 | 79,368.3 | 153.83 | 28.9 |
| 22 | 1.59 | 7315.9 | 15.07 | 4.0 |
| 23 | 1.59 | 22,119.1 | 45.65 | 4.5 |
| 24 | 1.41 | 57,912.5 | 151.65 | 19.6 |
| 25 | 1.52 | 22,139.9 | 50.08 | 3.8 |
| 26 | 0.93 | 20,068.7 | 119.89 | 12.7 |
| 27 | 0.94 | 8690.8 | 50.76 | 5.5 |
| 28 | 1.27 | 20,809.8 | 67.36 | 8.1 |
| 29 | 1.10 | 25,451.7 | 109.37 | 15.3 |
| 30 | 1.08 | 21,434.3 | 95.46 | 40.3 |
| 31 | 1.05 | 34,229.6 | 161.90 | 26.2 |
| 32 | 1.00 | 14,104.7 | 73.13 | 16.1 |
| 33 | 1.26 | 26,953.8 | 88.63 | 31.5 |
| 34 | 0.82 | 69,841.7 | 537.65 | 66.1 |
| 35 | 1.12 | 3954.4 | 16.28 | 4.7 |
| 36 | 1.45 | 23,479.8 | 58.22 | 9.1 |
| 37 | 0.89 | 20,517.2 | 133.50 | 30.9 |
| 38 | 0.49 | 26,340.3 | 569.67 | 19.0 |
| 39 | 1.05 | 104,106.2 | 490.07 | 14.9 |
| 40 | 1.26 | 6612.4 | 21.79 | 2.8 |
| 41 | 0.81 | 1688.9 | 13.52 | 3.4 |
| 42 | 0.97 | 12,563.8 | 68.93 | 7.9 |
| 43 | 1.07 | 11,667.0 | 53.55 | 7.8 |
| 44 | 1.00 | 22,069.4 | 115.37 | 14.8 |
| 45 | 1.04 | 7342.0 | 35.48 | 7.4 |

**Table A1.** *Cont.*

| Pyrome | Frequency Multiplier | Model-Predicted Expected Total | | |
|---|---|---|---|---|
| ID | (Same Cost) | Burned Area (ha/yr) | Runtime (s/yr) | Fires (yr$^{-1}$) |
| 46 | 1.18 | 27,092.5 | 100.58 | 9.7 |
| 47 | 1.31 | 21,751.2 | 66.38 | 4.2 |
| 48 | 0.82 | 4397.1 | 34.08 | 3.7 |
| 49 | 0.91 | 21,746.4 | 136.82 | 12.6 |
| 50 | 1.52 | 13,686.4 | 30.77 | 5.2 |
| 51 | 1.98 | 14,818.8 | 19.67 | 5.7 |
| 52 | 1.54 | 5911.9 | 13.02 | 2.8 |
| 53 | 1.41 | 50,181.2 | 131.02 | 23.7 |
| 54 | 1.32 | 23,050.1 | 69.03 | 12.9 |
| 55 | 1.34 | 45,158.7 | 130.02 | 48.8 |
| 56 | 1.36 | 9767.5 | 27.46 | 11.8 |
| 57 | 1.21 | 21,448.6 | 75.86 | 11.9 |
| 58 | 1.45 | 8336.1 | 20.74 | 4.9 |
| 59 | 1.21 | 12,332.9 | 43.69 | 9.3 |
| 60 | 1.04 | 5640.0 | 27.25 | 6.9 |
| 61 | 1.08 | 11,456.3 | 51.21 | 14.2 |
| 62 | 1.28 | 1152.2 | 3.67 | 3.2 |
| 63 | 1.15 | 1894.4 | 7.40 | 25.2 |
| 64 | 0.73 | 5974.9 | 58.92 | 53.3 |
| 65 | 1.14 | 7788.4 | 31.15 | 4.4 |
| 66 | 1.32 | 18,366.4 | 54.87 | 13.2 |
| 67 | 1.47 | 54,847.9 | 132.00 | 35.1 |
| 68 | 1.17 | 9244.3 | 35.42 | 15.6 |
| 69 | 1.08 | 17,408.7 | 77.19 | 17.9 |
| 70 | 1.11 | 22,850.1 | 96.64 | 21.1 |
| 71 | 1.06 | 13,200.3 | 61.42 | 9.4 |
| 72 | 0.90 | 35,760.5 | 227.94 | 60.4 |
| 73 | 1.05 | 20,397.4 | 97.16 | 15.7 |
| 74 | 1.72 | 11,097.9 | 19.48 | 6.2 |
| 75 | 1.91 | 8693.3 | 12.47 | 6.3 |
| 76 | 0.50 | 7261.7 | 151.96 | 18.0 |
| 77 | 1.13 | 49.6 | 0.20 | 0.0 |
| 78 | 1.17 | 6659.3 | 25.22 | 4.2 |
| 79 | 1.16 | 2682.5 | 10.45 | 6.8 |
| 80 | 0.98 | 7371.4 | 39.82 | 16.5 |
| 81 | 0.98 | 6986.1 | 37.50 | 42.5 |
| 82 | 1.20 | 10,818.0 | 39.03 | 26.2 |
| 83 | 1.01 | 6063.0 | 30.89 | 48.5 |
| 84 | 0.96 | 3881.1 | 21.79 | 20.7 |
| 85 | 1.06 | 5546.6 | 25.64 | 45.9 |
| 86 | 1.06 | 12,495.0 | 57.82 | 79.6 |
| 87 | 0.97 | 13,980.1 | 77.81 | 67.5 |
| 88 | 0.96 | 6212.0 | 34.99 | 21.8 |
| 89 | 1.04 | 11,715.3 | 56.52 | 14.3 |
| 90 | 2.06 | 5425.1 | 6.67 | 2.3 |
| 91 | 1.08 | 3879.9 | 17.43 | 65.4 |
| 92 | 0.97 | 2469.0 | 13.58 | 34.5 |
| 93 | 0.97 | 16,873.5 | 93.54 | 627.9 |
| 94 | 1.00 | 3696.6 | 19.13 | 46.7 |
| 95 | 0.40 | 14,253.1 | 475.06 | 69.9 |
| 96 | 0.39 | 1101.0 | 38.10 | 2.1 |
| 97 | 0.75 | 7047.1 | 65.41 | 65.5 |
| 98 | 1.04 | 964.2 | 4.67 | 20.5 |

**Table A1.** *Cont.*

| Pyrome | Frequency Multiplier | Model-Predicted Expected Total | | |
| | | Burned Area (ha/yr) | Runtime (s/yr) | Fires (yr$^{-1}$) |
| ID | (Same Cost) | | | |
|---|---|---|---|---|
| 99 | 1.06 | 680.5 | 3.18 | 14.5 |
| 100 | 1.02 | 1316.2 | 6.58 | 15.7 |
| 101 | 0.79 | 793.6 | 6.69 | 7.7 |
| 102 | 1.02 | 544.3 | 2.73 | 15.7 |
| 103 | 0.81 | 107.8 | 0.85 | 10.1 |
| 104 | 0.77 | 130.7 | 1.15 | 10.0 |
| 105 | 0.94 | 146.1 | 0.87 | 5.4 |
| 106 | 0.95 | 4321.1 | 24.81 | 91.9 |
| 107 | 1.06 | 2006.8 | 9.22 | 22.8 |
| 108 | 1.11 | 141.3 | 0.60 | 2.0 |
| 109 | 0.92 | 21,235.4 | 130.98 | 137.4 |
| 110 | 1.11 | 8691.4 | 36.47 | 103.1 |
| 111 | 0.91 | 13,278.6 | 83.02 | 114.6 |
| 112 | 0.86 | 57,716.8 | 409.04 | 313.4 |
| 113 | 1.04 | 52,045.2 | 252.31 | 45.0 |
| 114 | 0.94 | 1905.3 | 11.24 | 24.4 |
| 115 | 0.98 | 15,348.7 | 84.02 | 123.8 |
| 116 | 1.00 | 2356.0 | 12.37 | 23.7 |
| 117 | 0.95 | 1816.4 | 10.42 | 14.0 |
| 118 | 0.97 | 3994.9 | 22.06 | 80.9 |
| 119 | 1.04 | 3806.6 | 18.19 | 26.6 |
| 120 | 1.02 | 11,617.3 | 58.66 | 174.3 |
| 121 | 0.98 | 8300.3 | 45.44 | 199.3 |
| 122 | 0.96 | 1127.7 | 6.42 | 21.4 |
| 123 | 0.98 | 236.2 | 1.29 | 5.4 |
| 124 | 0.91 | 916.8 | 5.75 | 18.3 |
| 125 | 1.02 | 309.1 | 1.56 | 9.9 |
| 126 | 1.01 | 6965.1 | 35.50 | 152.1 |
| 127 | 1.08 | 12,631.8 | 56.76 | 3.8 |
| 128 | 1.34 | 11,486.8 | 33.47 | 11.8 |

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
