# Peer review of "Importance Sampling for Cost-Optimized Estimation of Burn Probability Maps in Wildfire Monte Carlo Simulations"

_fire, doi:10.3390/fire7120455_

Round 1
Reviewer 1 Report
Comments and Suggestions for Authors
Importance Sampling for cost-optimized estimation of burn-probability maps in wildfire Monte Carlo simulations
This approach fits into the broader framework of cost-optimized estimation methods, where computational resources are allocated in a way that maximizes the accuracy of the desired output (burn-probability maps) while minimizing simulation time and effort.
1). Importance sampling requires careful implementation, including proper weighting of the samples and validation to ensure that the results are unbiased and representative of actual wildfire dynamics. The authors need to mention that process and main contributions in the Introduction.
2). The related work section was skipped in this paper. All scientific journal papers should follow generally accepted structures. Strongly suggest making a related works section to review and mention recently published works similar to submitted papers based on ML and DL and etc.
3). I'm wondering what kinds of tools or software they used to implement difficult calculations. Please make a table to give more detailed information about employed hardware and software in this research and implementations.
4). Figure 2 requires specific explanations. The authors saying "More details in table 2" but it's not mentioned what is color differences and which color warns of fire accidents. Authors should understand, that not only skilled researchers read the papers, there are some new researchers who may be confused due to the complex writing style.
5). Authors approach to write abstract and conclusion sections not clearly. For instance, the Abstract looks as describe the contents of the thesis work which should be end of the Introduction section and the Conclusion part looks continue of the discussion section. It should be revised and shortly should mention the main achievements, faced difficulties and used approaches to fix those issues.
6). I think authors should improve their scientific writing style and skills even if interesting work is introduced, many readers may ignore some research papers due to not understanding the complexity and unclear structures.
Comments on the Quality of English Language
can be improved
Reviewer 2 Report
Comments and Suggestions for Authors
fire-3276739 – Importance Sampling for cost-optimized estimation of burn-probability maps in wildfire Monte Carlo simulations - A Review
I have now completed my evaluation of the manuscript fire-3276739. In this study the authors propose using importance sampling to optimize Monte Carlo simulations for wildfires. While the topic is of interest to the wildfire simulations community and the authors have done a good job overall, there are several important issues that must be addressed before the paper is ready for publication in Fire. In the following paragraphs I will describe both the major issues and specific comments the authors should address.
Line 48 – typo (willàwith)
The writing frequently moves from present to future – both are acceptable but please ensure consistency.
The authors should provide details regarding the FireFactor project (reference 12). I agree that they do not need to perform experiments to prove its validity, but some details regarding their simulations including the choice of hyperparameters and weather data must be provided to understand the research.
Also, please provide descriptive statistics of the historical data described in the research.
Line 362 – the authors should provide justification for this choice.
Table 2 – this is far too much data to be presented in the manuscript. Please move to appendix and provide summary statistics instead of the full table, while emphasizing the relevant conclusions.
Section 3.4. – this is the first time Pyrome 33 is mentioned. The authors should describe the research design much earlier, and then present its results. Also provide some details about it and why it was chosen.
The Discussion and Conclusions are written in a very uncommon way. Currently the Discussion section only discusses one limitation of the study, and the Conclusions section summarizes the research. Please expand the Discussion section with a meaningful discussion of the results of the research, engaging with the current literature. Also, I would not itemize the Conclusions section.
The number of references also reflects the scarce literature review and discussion of the paper’s possible implications. The authors should further expand on the motivation for this study (How common are Monte Carlo simulations in this field? Are they the only method or are other techniques, such as Machine Learning, used for wildfire prediction? What do studies usually predict – burned areas, spread rate? etc.). Also, they should discuss the possible impact of their research on this literature.
As I previously stated, I think the paper has great potential and I encourage the authors to improve it based on these guidelines.
Round 2
Reviewer 1 Report
Comments and Suggestions for Authors
Comments for second round to improve paper quality!
1). At the end of the Introduction, it is necessary to add the main contributions and a description of the further structure of the manuscript.
2). And Related Work section should be separated from the Introduction as "2. Related Work" and can be extended with more similar published works between 2023-2024.
3). Materials and Methods. I suggest making a general flowchart of the proposed work to clearly and briefly explain their steps.
4). Materials and Methods. They included too many equations and math expressions to show the simulations. Please confirm all of them necessary. Because modern IT tools can calculate all equations within one function.
5). Please try to discuss some literature in your Discussion section that can be used to detect visible fire scenes in your future direction.
Comments on the Quality of English Languageok
Reviewer 2 Report
Comments and Suggestions for Authors
fire-3276739 R1 – Importance Sampling for cost-optimized estimation of burn-probability maps in wildfire Monte Carlo simulations - A Review
I have gone over the revised version of the manuscript. I believe the current version is much more mature, the authors have done a very good job in addressing the previous issues, and the manuscript is now ready for publication. I congratulate the authors on their accomplishments.
Author Response
The authors thank the reviewer for the kind words.